# Convergence Properties of Deep Neural Networks on Separable Data

## Abstract

While a lot of progress has been made in recent years, the dynamics of learning in deep nonlinear neural networks remain to this day largely misunderstood. In this work, we study the case of binary classification and prove various properties of learning in such networks under strong assumptions such as linear separability of the data. Extending existing results from the linear case, we confirm empirical observations by proving that the classification error also follows a sigmoidal shape in nonlinear architectures. We show that given proper initialization, learning expounds parallel independent modes and that certain regions of parameter space might lead to failed training. We also demonstrate that input norm and features' frequency in the dataset lead to distinct convergence speeds which might shed some light on the generalization capabilities of deep neural networks. We provide a comparison between the dynamics of learning with cross-entropy and hinge losses, which could prove useful to understand recent progress in the training of generative adversarial networks. Finally, we identify a phenomenon that we baptize *gradient starvation* where the most frequent features in a dataset prevent the learning of other less frequent but equally informative features.

## 1 Introduction

Due to extremely complex interactions between millions of parameters, nonlinear activation functions and optimization techniques, the dynamics of learning observed in deep neural networks remain much of a mystery to this day. What principles govern the evolution of the neural network weights? Why does the training error evolve as it does? How do data and optimization techniques like stochastic gradient descent interact? Where does the *implicit regularization* of deep neural networks trained with stochastic gradient descent come from? Shedding some light on those questions would make training neural networks more understandable, and potentially pave the way to better techniques.

It is commonly accepted that learning is composed of alternating phases: plateaus where the error remains fairly constant and periods of fast improvement where a lot of progress is made over the course of few epochs (Saxe et al., 2013a). Theoretic explanations of that phenomenon exist in the case of regression on linear neural networks (Saxe, 2015) but extensions to the nonlinear case (Heskes & Kappen, 1993; Raghu et al., 2017; Arora et al., 2018) fail to provide analytical solutions.

It has been observed in countless experiments that deep networks present strong generalization abilities. Those abilities are however difficult to ground in solid theoretical foundations. The fact that deep network have millions of parameters – a number sometimes orders of magnitude larger than the dataset size – contradicts the expectations set by classic statistical learning theory on the necessity of regularizers (Vapnik, 1998; Poggio et al., 2004). This observation drove Zhang et al. (2016) to suggest the existence of an *implicit regularization* happening during the training of deep neural networks. Advani & Saxe (2017) show that the dynamics of gradient descent can protect against overfitting in large networks. Kleinberg et al. (2018) also offer some explanations of the phenomenon but understanding its roots remains an open problem.

In this work, we study the learning dynamics of a deep nonlinear neural network – *i.e.* how its weights and outputs evolve throughout learning – trained on a standard classification task using two different losses: the cross-entropy and the hinge loss. We mainly focus on binary classification, some of the results and properties can however be extended to the multi-class case. The questions we address in Sections 3, 4 and 5 respectively can be summarized as follows:

*How does the confidence of a classifier evolve throughout learning?*
*How does the loss used during training impact its dynamics?*
*Which properties of the features present in a dataset impact learning, and how?*

**Independent mode learning** We show that, similarly to the case of linear networks and under certain initial conditions, learning happens independently between different classes, *i.e.* classes induce a partition of the network activations, corresponding to orthogonal modes of the data.

**Learning dynamics** We prove that in accordance to experimental findings, the hidden activations and the classification error of the network show a sigmoidal shape with slow learning at the beginning followed by fast saturation of the curve. We also characterize a region in the initialization space where learning is frozen or eventually dies out.

**Hinge loss** We study how using the hinge loss impacts learning and quantitatively compare it to the classic cross-entropy loss. We show that the hinge loss allows one to solve a classification task much faster, by providing strong gradients no matter how close to convergence the neural network is.

**Gradient starvation** Finally, we identify a phenomenon that we call *gradient starvation* where the most frequent features present in the dataset *starve* the learning of other very informative but less frequent features. Gradient starvation occurs naturally when training a neural network with gradient descent and might be part of the explanation as to why neural networks generalize so well. They intrinsically implement a variant of Occam's razor (Ariew, 1976): *the simplest explanation is the one they converge to first*.

## 2 SETUP AND NOTATIONS

We are interested in a simple binary classification task, solved by training a deep neural network with gradient descent. This simple setup encompasses for instance the training of generative adversarial networks discriminators. Some of our results extend to multi-class classification, but, for the sake of conciseness, that case is treated in Appendix A. We let $D = \{(x_i, l_i)\}_{1 \leq i \leq n} \subset \mathbb{R}^d \times \{1, 2\}$ denote our dataset of vectors and labels. The classifier we consider is a simple neural network with one hidden layer of $h$ neurons and a ReLU non-linearity (see Fig. 1). The output of the network is passed through a function denoted $o$ which is either the sigmoid $\sigma$ in the binary cross-entropy case or the identity in the hinge loss case. The full function can be written as

$$P_t(x) := o(u_t(x)) := o(Z_t^T (W_t x)_+),$$

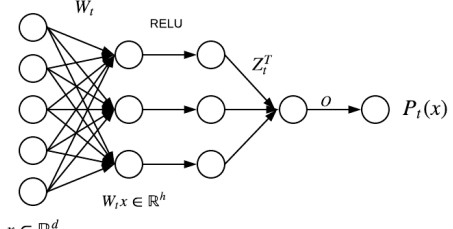

where $W_t$ is an $h \times d$ matrix and $Z_t$ a vector of length $h$. In the $C$-class case, $Z_t$ is instead a $C \times h$ matrix and $o$ the softmax function. An extension to deeper networks can be found in Appendix B. $W_t$ and $Z_t$ are the parameters of the neural network. The subscript $+$ (resp. $t$) denotes the positive part of a real number (resp. the state of the element at time step $t$ of training).

Figure 1: Network Architecture

The superscript $T$ stands for the transpose operation. In the case of cross-entropy, $P_t(x)$ represents the probability that $x$ belongs to class 1, for the hinge loss, $P_t(x)$ is trained to reach $\{1, -1\}$ for classes 1 and 2. For a given class $k$, we let $D_k$ denote the set of vectors belonging to it. We make the assumption

(H1) For any $x, x' \in D_k$, $x^T x' > 0$. For any $x \in D_1$ and $x' \in D_2$, $x^T x' \leq 0$.

It implies linear separability of the data, an assumption often necessary in theoretical studies (Soudry et al., 2017; Liao & Couillet, 2018; Nacson et al., 2018; Xu et al., 2018) and the positioning of the origin between the two sets. It is a very strong assumption which admittedly bypasses a large part of the deep learning dynamics. Nevertheless, it allows the discovery of interesting properties and is potentially a first step towards understanding behaviors observed in more general settings.

# 3 LEARNING DYNAMICS FOR BINARY CROSS-ENTROPY

In this section, we focus on the case of the binary cross-entropy loss

$$L_{BCE}(W_t, Z_t; x) = -\mathbb{1}_{x \in D_1} \log(\sigma(Z_t^T(W_t x)_+)) - \mathbb{1}_{x \in D_2} \log(1 - \sigma(Z_t^T(W_t x)_+)),$$

and train our network using stochastic gradient descent to minimize $L_{BCE}$.

## 3.1 INDEPENDENT MODES OF LEARNING

Our first lemma states that over the course of training and under suitable initialization, the active neurons of the hidden layer remain the same for each datapoint, and the coordinates of $Z_t$ remain of the same sign. To prove it, we let $w_t^i$ denote the $i$-th row of $W_t$ and make the additional assumptions: there exists a partition $\{\mathcal{I}_1, \mathcal{I}_2\}$ of $\{1, \ldots, h\}$ such that with $k \in \{1, 2\}$

> (H2) For any $i \in \mathcal{I}_k$, $x \in D_k$ and $x' \notin D_k$, $w_0^i x > 0$ and $w_0^i x' \leq 0$.

> (H3) The $i$-th coordinate of $Z_0$ is positive if $i \in \mathcal{I}_1$, negative otherwise.

Assumption (H2) states that at the beginning of training, data points from different classes do not activate the same neurons. It is an analogue to the orthogonal initialization used in Saxe et al. (2013b). In Appendix A.8, we show that relaxing it hints towards an extended period of slow learning in the early stages of training. (H3) is introduced for Lemma 3.1 and Theorem 3.2 but will be relaxed later.

**Lemma 3.1.** *For any $k \in \{1, 2\}$, $x \in D_k$ and $t \geq 0$, the only non-negative elements of $W_t x$ are the ones with an index $i \in \mathcal{I}_k$. The signs of the coordinates of $Z_t$ remain the same throughout training.*

This lemma proves that updates to the parameters of our network are fully decoupled from one class to the other. An update for a data point in $D_k$ will only influence the corresponding *active* rows and elements of $W_t$ and $Z_t$. This "independent mode learning" is an equivalent of the results by Saxe et al. (2013b) in a non-linear network trained on the cross entropy loss. The proof of the lemma and its extension to $N - 1$ hidden layers and multi-class classification can be found in Appendix A.

## 3.2 LEARNING DYNAMICS

We are now interested in the actual dynamics of learning, and move from discrete updates to continuous ones by considering an infinitesimal learning rate $\alpha$ (Heskes & Kappen, 1993). Lemma 3.1 can easily be extended to this setting. For simplicity we assume $h = 2$, but similar results hold for arbitrary $h$ (see Appendix A.4). For the moment, we maintain the assumptions (H1-3).

**Theorem 3.2.** *Assuming that each class $k$ contains the same vector $x_k$ repeated $|D_k|$ times, then the output of the classifier on $D_k$ verifies (with $p_k = |D_k|/|D|$ the fraction of $D$ belonging to $D_k$):*

$$P_t(x_k \in D_k) = \sigma(u(\|x_k\| p_k t)),$$

*where $u$ is defined below. The classification curves are sigmoidal and can be found on Fig. 2 Right.*

*Proof.* To simplify the notations, we arbitrarily assume that $\mathcal{I}_1 = \{1\}$ and we write $w_t$ and $z_t$ the row and element modified by an update made using $x \in D_1$ (the case of $D_2$ can be treated symmetrically). By the independence above, we know that $w_t$ and $z_t$ are only affected by updates from $D_1$. This greatly simplifies our evolution equations to: $w_t' = \delta_f(x) z_t \, x^T$ and $z_t' = \delta_f(x) \, w_t x$ where $\delta_f(x)$ is the gradient of the loss with respect to the pre-sigmoid output of the network $u_t(x)$: $\delta_f(x) = \mathbb{1}_{\{k=1\}} - \sigma(u_t(x))$ and the prime indicates a time derivative. We let $y_t = w_t x$, which gives

$$y_t' = \frac{x^T x \, z_t}{1 + e^{y_t z_t}}, \qquad\qquad z_t' = \frac{y_t}{1 + e^{y_t z_t}}. \qquad (1)$$

Writing $\|x\|^2 = x^T x$, we see that the quantity $y_t^2 - \|x\|^2 z_t^2$ is an invariant of the problem, so its solutions live on hyperbolas of equation $y^2 - \|x\|^2 z^2 = \pm c$ with $c := |y_0^2 - \|x\|^2 z_0^2|$.

We only treat the case of a degenerate hyperbola $c = 0$ *i.e.* $y_0^2 = \|x\|^2 z_0^2$, and refer the interested reader to Appendix A.3 for the full derivation. In the case $c = 0$, we have $\forall t, y_t = \|x\| z_t$ (those quantities are both positive as $x \in D_1$ and (H2-3)). $u(t) := z_t w_t x = z_t y_t$ thus follows the equation

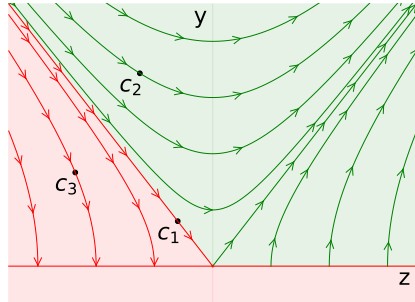 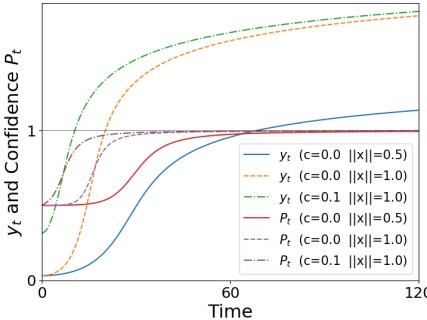

Figure 2: *Left.* Phase diagram representing the dynamics of learning for the couple $(z_t, y_t)$ depending on its initialization. $y_t$ is the value for the class considered, in which all examples have lined up. Each couple lives on a hyperbola. The slope of the linear curves is equal $\pm\|x\|$ (set to $0.7$ in this diagram). The green region represents the initializations of $(y, z)$ where the classification task will be solved by the network. In the red region, learning does not start (the neuron is inactive at the beginning of training) or collapses as the neuron dies off when $y$ reaches $0$. The $c_i$ points show the three cases from Section 3.3. *Right.* $y_t$ and $P_t(x \in D_1) = \sigma(z_t y_t)$ for different values of $c$ and $\|x\|$.

$u'(t) = 2\|x\|u(t)\sigma(-u(t))$. One can see the equivalence between our evolution equation and Eq. (10) in Saxe et al. (2013b). Its analytical solution is (see Appendix A.3):

$$u(t) = (\log + Ei)^{<-1>}(2\|x\|t + \log(u_0) + Ei(u_0)), \tag{2}$$

where $Ei$ is the exponential integral (Wiki., 2018) and $<-1>$ denotes the inverse function.

We let $\bar{u}$ denote that function for $\|x\| = 1$. For $x \in D_2$, the degeneracy assumption becomes $y_0 = -\|x\|z_0$. It can be shown similarly that $v(t) := z_t y_t$ verifies the equation $v'(t) = 2\|x\|v(t)\sigma(v(t))$ with a negative initial condition (H2-3). In other words, $u$ and $v$ follow symmetric trajectories on the positive/negative real line. Below, $\bar{u}$ (resp. $\bar{v}$) denote those two trajectories for $\|x\| = 1$ and initial conditions $u_0 > 0$ (resp. $v_0 < 0$). Let us now write $p_1 = |D_1|/|D|$ the fraction of points belonging to $D_1$. Because we sample randomly from the dataset, this amounts to sampling $p_1$ (resp. $1 - p_1$) points from $D_1$ (resp. $D_2$) for each time unit during training, *i.e.* to rescaling the time axis by $p_1$ for $D_1$ and $1 - p_1$ for $D_2$. This allows us to quantify the network's performance at any time $t$:

$$\begin{cases} P_t(x \in D_1) &= \sigma(\bar{u}(\|x\|p_1 t)) & \text{if } x \in D_1 \\ P_t(x \in D_2) &= \sigma(-\bar{v}(\|x\|(1 - p_1)t)) & \text{if } x \in D_2 \end{cases} \tag{3}$$

In particular, the convergence of $u(t)$ to $+\infty$ can be bounded using our results: convergence happens at a rate slower than $\log(t)$ (Appendix A.3), a fact proved on its own by Soudry et al. (2017). □

**Interpretation** Fig. 2 *Right.* shows the learning dynamics for different values of $\|x\|$ and $c$. One common characteristic between all the curves is their sigmoidal shape. Learning is slow at first, then accelerates before saturating. This is aligned with empirical results from the literature. We also see on *e.g.* the blue and yellow curves that a larger $\|x\|$ (or similarly a larger $p$) converges much faster. The effect of $c$ on the dynamics can mostly been seen at the beginning of training (for instance on the green and yellow curves). It fades as convergence happens, corresponding to points of the hyperbolas getting closer to the asymptote $y = \|x\|z$, see Fig. 2 *Left.* and below for more details. We can characterize the convergence speeds more quantitatively with the following corollary.

**Corollary 3.3.** *Let $\delta$ be the required accuracy on the classification task (i.e. $P_t(x \in D_1) \geq 1 - \delta$ for $x \in D_1$). Under certain assumptions, the times $t_1^*$ and $t_2^*$ required to reach that accuracy for each classifier verify $\frac{t_2^*}{t_1^*} \approx \frac{\|x_1\|}{\|x_2\|}\frac{p_1}{1 - p_1}$ where $\|x_k\|$ is the norm of vector $\|x_k\|$ from class $k$.*

The proof can be found in Appendix A.5. More frequent classes and larger inputs will be classified at a given level of confidence faster. The class frequency observation is fairly straightforward as updates on a more frequent class occur at a higher rate. As far as input sizes are considered, this can be seen as an analogous to the results from Saxe et al. (2013b) stating that input-output correlations

drive the speed of learning. Because a sigmoid is applied on the network output, its (pre-sigmoid) targets are sent to $\pm\infty$. A larger input is more correlated with its target and converges faster.

**On the assumptions** The assumption that each class only contains one vector allows us to obtain *the first closed-form solutions of the learning dynamics for the binary cross-entropy*. It can be relaxed to classes containing orthogonal datapoints (see Appendix A.7) which still remains restrictive. A possible interpretation is the following: if one were to consider a deep neural network that has learnt two discriminative features for the two classes, applying classic SGD on those features would result in a learning rate proportional to the prominence of those two features in the original dataset, and to learning curves of that exact shape. It is worth noting that such shapes are regularly observed by ML practitioners (Saxe et al., 2013b), our results reveal insights - otherwise unobtainable - into them.

### 3.3 Phase diagram

In this section, we build the phase diagram of Fig. 2 *Left*. The notations follow Theorem 3.2, in particular $y_t = w_t x$. So far, we have considered points in the top-right quadrant. In that region, the couple $(z_t, y_t)$ lives on a hyperbola of equation $y^2 - \|x\|^2 z^2 = \pm c$ where $c \geq 0$. The sign in the equation is defined by the position of $(z_0, y_0)$ relative to the function $y = \|x\|z$ (positive if above, negative otherwise). If $(z_0, y_0)$ is originally on that line, it will remain there throughout training. We now explore the rest of the parameter space by relaxing some of our assumptions. We still consider a point $x \in D_1$, the diagram for $D_2$ can be obtained by mirroring the $z$ axis.

*Assumption (H2).* Let us first consider the simple case of $w_0 x = y_0 < 0$. The neuron is initially inactive because of the ReLU. No updates will ever be made to $w_t$ during training. This corresponds to the bottom half of the phase diagram, the parameters are frozen (also see Advani & Saxe (2017)).

*Assumption (H3).* We now assume that $z_0 \leq 0$. In that case, a simple extension of Lemma 3.1 shows that learning still happens independently on each row of $w_t$. The outcome from Theorem 3.2 is still valid: the couple $(z_t, y_t)$ lives on a hyperbola. It is however not guaranteed anymore that $y_t$ shall remain positive throughout training. There are three possible situations (numbered 1 to 3), each represented by the corresponding point on the diagram.

1) If $y_0 = -\|x\|z_0$, then the points $(z_t, y_t)$ are stuck in the top-left quadrant and converge to zero. The equation verified by the logit $u(t)$ is $u'(t) = -2\|x\|u(t)\sigma(-u(t))$ (see Appendix A.6).

2) If $y_0 > -\|x\|z_0$, the points $(z_t, y_t)$ move on the hyperbola towards the top-right quadrant, at which point $z_t$ becomes positive and $y_t$ starts increasing again.

3) If $y_0 < -\|x\|z_0$, the points $(z_t, y_t)$ move on the hyperbola towards the bottom-left quadrant, at which point $y_t$ becomes negative. When that happens, the neuron dies out, learning stops.

Only in the second case will the classifier end up solving the task: random initialization only functions in certain parts of the $(y_t, z_t)$ space. Those findings are summarized in the phase diagram Fig. 2 *Left*. The red region represents the initialization where the network will not be able to solve the task.

**Failure modes** Assumption (H2) essentially means that the network's first layer is able to separate the data at $t = 0$. It is remarkable that even under (H2) the model fails on a non-zero measure set of the initialization space (top-left red region of Fig. 2 *Left*): in that region, it converges to a classifier assigning a probability of $0.5$ to the true class (dash-dotted curves in Fig. 3 *Right*).

### 3.4 Relaxing Assumption (H2)

Assumption (H2) states that for any $i \in \mathcal{I}_k$, $x \in D_k$ and $x' \notin D_k$, $w_0^i x > 0$ and $w_0^i x' \leq 0$. We now relax it by assuming that a point $x_2$ in $D_2$ verifies $w_0^1 x_2 > 0$ and we study the evolution of $w_t^1$. We consider updates coming from sampling equally $x_1$ from $D_1$ and $x_2$ from $D_2$. To simplify the analysis, we assume that $x_1^T x_2 = 0$ and write $\alpha_t = w_t^1 x_1$ (equivalent to $y_t$ above) and $\beta_t = w_t^1 x_2$. The triplet $(\alpha_t, \beta_t, z_t)$ satisfies the following system of ODEs (see Appendix A.8 for the derivation):

$$\alpha'_t = \frac{\|x_1\|^2 z_t}{1 + e^{z_t \alpha_t}}, \qquad \beta'_t = -\frac{\|x_2\|^2 z_t}{1 + e^{-z_t \beta_t}}, \qquad z'_t = \frac{\alpha_t}{1 + e^{z_t \alpha_t}} - \frac{\beta_t}{1 + e^{-z_t \beta_t}}. \qquad (4)$$

From our relaxation of (H2), we have $\alpha_0, \beta_0 > 0$. Since the system is symmetric under the transformation $(\alpha_t, \beta_t, z_t) \rightarrow (\beta_t, \alpha_t, -z_t)$, we can assume $z_0 \geq 0$. Due to the ReLU activations

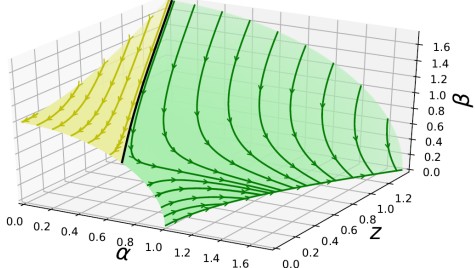 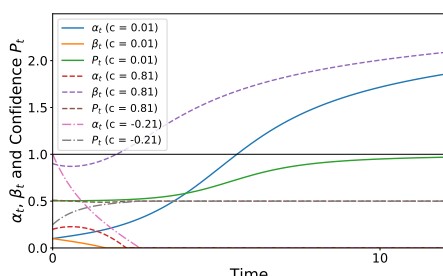

Figure 3: *Left.* Solutions of (4) for different initializations and $c = 1$. *Right.* Values of $\alpha_t$, $\beta_t$ and $P_t$ the confidence of the classifier on an example from class $D_1$ for three different initializations. The "full" curves correspond to $(\alpha_0, \beta_0, z_0) = (0.1, 0.1, 0.1)$ *i.e.* a trajectory in the green region where $\beta_t$ reaches 0 (orange curve). The confidence on class $D_1$ tends to 1 (green curve). The "dashed" curves correspond to $(\alpha_0, \beta_0, z_0) = (0.2, 0.9, 0.2)$ *i.e.* a trajectory in the yellow region, corresponding to $\alpha_t$ reaching 0 (red curve). The confidence on $D_1$ goes to 0.5 in that case (brown curve), and the confidence on class $D_2$ goes to 1 (not shown). The "dash-dotted" curves correspond to $(\alpha_0, \beta_0, z_0) = (1, 0, -1.1)$ and are an instance of the aforementioned failure mode: $\alpha_t$ (or equivalently $y_t$) tends to 0 (pink curve), $\beta_t$ (not shown) remains 0 and $P_t$ tends to 0.5 (grey curve).

of the network, whenever $\alpha_t$ or $\beta_t$ reaches zero, it becomes constant and its contribution to $z_t$ disappears: the model reaches the independent modes of learning regime from Section 3.1 and evolves according to the results above (*e.g.* green hyperbolas in the plane $\beta = 0$ on Fig. 3 *Left*). If $\beta_t$ (resp. $\alpha_t$) reaches 0 at some point, $D_1$ (resp. $D_2$) becomes the only class activating the neuron. Let us now characterize how the initialization of the network influences the outcome of learning.

**Theorem 3.4.** *Letting $c := \alpha_0^2/\|x_1\|^2 + \beta_0^2/\|x_2\|^2 - z_0^2$, the solutions of (4) verify for all $t \geq 0$, $\alpha_t^2/\|x_1\|^2 + \beta_t^2/\|x_2\|^2 - z_t^2 = c$. In other terms, they live on hyperboloids (see Fig. 3 Left).*

*If $c \leq 0$, $\beta_t$ reaches 0 at some point during training (Fig. 6 of the Appendix). If $c > 0$, there exists a curve $\mathcal{C}_c$ (shown in black on Fig. 3 Left) such that as $t \to +\infty$, for any initialization $(\alpha_0, \beta_0, z_0) \in \mathcal{C}_c$:*

$$\alpha_t \to (\frac{1}{\|x_1\|^2} + \frac{1}{\|x_2\|^2})^{-1/2}\sqrt{c}, \qquad \beta_t \to (\frac{1}{\|x_1\|^2} + \frac{1}{\|x_2\|^2})^{-1/2}\sqrt{c}, \qquad z_t \to 0.$$

*That curve defines two regions of the initialization space. In one, colored yellow on Fig. 3 Left, the trajectories verify $\alpha_t = 0$ for some $t$. In the other, colored green, $\beta_t$ reaches 0 at some point.*

**Interpretation** The proof of the theorem can be found in Appendix A.8. Fig. 3 *Left* shows some solutions of (4). Concretely, we see from the equations that the sign of $z_t$ determines whether $\alpha_t$ and $\beta_t$ increase or decrease, and how fast they do so. $z_t$'s evolution on the other hand is the result of a competition between $\alpha_t$ and $\beta_t$. If $z_0$ and/or $\alpha_0$ are sufficiently large, $\beta_t$ will decrease fast and long enough to reach 0 at some point (green curves). Conversely, for a large $\beta_0$, $z_t$ can reach 0 before $\beta_t$. When that is the case, $\alpha_t$ then decreases until it reaches 0 (yellow curves). We plot examples of those behaviors in Fig. 3 *Right*. We notice in particular the classic sigmoidal shape appearing, even when Assumption (H2) is violated. This can be explained as follows. In the regime of small initializations (customary in deep learning), the competition between $\alpha_t$, $\beta_t$ and $z_t$ happens in a part of parameter space where all the weights are small (*i.e.* where the confidence of the network is close to 0.5). When one class finally prevails over the other, *e.g.* $\beta_t$ reaching 0 (orange curve in the plot), the analytical solutions from previous sections apply and the sigmoidal shape arises.

# 4  ON THE HINGE LOSS

Recent results in the field of generative adversarial networks have resurrected the hinge loss (Miyato et al., 2018). While its exact impact on performance is unclear, we run a small experiment to show its ability to generate better samples than the customary cross-entropy (see Fig. 4 and Appendix E). In order to perhaps uncover reasons behind its efficiency, we extend our results to the hinge loss:

$$L_H(W_t, Z_t; x) = \max(0, 1 - Z_t^T(W_t x)_+ \cdot (\mathbb{1}_{x \in D_1} - \mathbb{1}_{x \in D_2})).$$

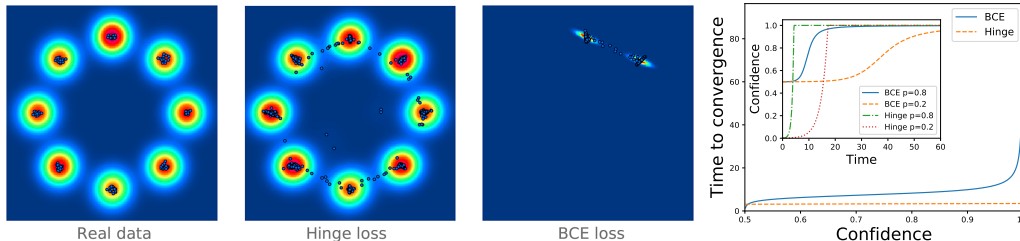

Figure 4: *Left*. The three figures on the left are the result of training a generative adversarial network on 8 Gaussians (see Appendix E for details on the experiment). The samples from the hinge loss are incomparably better. *Right*. Comparison between hinge loss and binary cross-entropy: training time required to reach a confidence $\delta$ on the classification problem. Subplot: Solutions of Eqs. 3 and 5.

The hinge loss is non differentiable, but one can simply consider that learning stops as soon as the output of the network reaches 1 (resp. -1) for class $D_1$ (resp. $D_2$). Under the same assumptions than in Theorem 3.2 (some of which can be relaxed, see Appendix C), we have the following result:

**Theorem 4.1.** *For $x \in D_1$ (for $D_2$ it is simply the opposite), the output $u(t)$ of the network verifies*

$$u(t) = \min(1, \ u_0 \ e^{2p\|x\|t}), \qquad u(t) = \min(1, \ \frac{c}{2\|x\|} \sinh(\theta_0 + 2p\|x\|t)), \qquad (5)$$

*where $\theta_0 = \cosh^{-1}(\frac{y_0^2 + \|x\|^2 z_0^2}{c})$ and the left and right equations correspond to $c = 0$ and $c \neq 0$.*

*Proof.* Simple computations show that the dynamics of the system are governed by $y'_t = x^T x \, z_t$ and $z'_t = y_t$. Following the method from Section 3, we see that $u'(t) = 2\|x\|u(t)$ in the case where $c = 0$, leading to to the result. When $c \neq 0$, a classic hyperbolic change of variables allows to find the solution. Its full derivation is presented in Appendix C. $\qquad\square$

The learning curves are plotted in Fig. 4 *Right*. We notice a *hard* sigmoidal shape corresponding to learning stopping when $u_t$ reaches 1. Confidence increases exponentially in $t$, much faster than for binary cross-entropy (all other parameters kept equal) where $u(t) \sim \log(t)$. With $\delta$ the required confidence for our classifier, the time $t^*$ required to reach $\delta$ can easily be computed. We plot it in Fig. 4 *Right* which confirms visually that the hinge loss converges much faster. We also notice the expected divergence of $t^*$ for the binary cross entropy as $\delta$ reaches 1 (training never converges in that case). We refer the interested reader to Appendix C for a more general treatment of the Hinge loss, which fully relaxes the assumption on the number of points in the classes.

## 5  GRADIENT STARVATION

In this section, we attempt to quantify the impact of feature frequency inside a given class. We keep our simplified framework and consider that the input $x$ to our network is composed of two underlying features $x_1 \in \mathbb{R}^{d_1}$ and $x_2 \in \mathbb{R}^{d_2}$ with $d = d_1 + d_2$. We let $(x_1, x_2) \in \mathbb{R}^d$ denote the concatenation of the vectors $x_1$ and $x_2$. We assume that all the points in class $D_1$ contain the feature $x_1$ but only a fraction $\lambda$ of them contains the feature $x_2$. This is equivalent to making continuous gradient updates using the vector $(x_1, x_2)$ with a rate $\lambda$ and the vector $(x_1, 0)$ with a rate $1 - \lambda$. We also assume that those features are fully informative for $D_1$ – *i.e.* are absent from class $D_2$. A network trained using gradient descent on the dataset we just described has the following property

*Even though the feature represented by $x_2$ is fully informative of the class, the network will not classify a sample containing only $x_2$ with high confidence.*

It is the result of a phenomenon we coin *gradient starvation* where the most frequent features starve the gradient for the least frequent ones, resulting in a slower learning of those:

**Theorem 5.1.** *Let $\delta$ be our confidence requirement on class $D_1$ i.e. training stops as soon as $\forall x \in D_1, \ P_t(x \in D_1) \geq 1 - \delta$, and let $t^*$ denote that instant. Then, under some mild assumptions,*

$$P_{t^*}((0, x_2) \in D_1) \leq \frac{1}{1 + e^{-\lambda \log(\frac{1-\delta}{\delta})}} \ . \qquad (6)$$

*Proof.* From Lemma 3.1, assumptions (H2-3) are sufficient to guarantee independent mode learning as well as positiveness of $z_t$. We decompose $w_t = (\alpha_t x_1, \beta_t x_2) + (x_1^\perp, x_2^\perp)$ where $x_1^T x_1^\perp = x_2^T x_2^\perp = 0$, and assume that $\alpha_0 \geq \beta_0/\lambda > 0$ (in App. D, we relax some of those assumptions and prove an equivalent result). The evolution equation for $w_t$ is $w_t' = \frac{z_t x}{1 + e^{z_t w_t x}}$ with $x = (x_1, x_2)$ (resp. $(x_1, 0)$) at an $\lambda$ (resp. $1 - \lambda$) rate. Projecting on $x_1$ and $x_2$ gives

$$\alpha_t' = \lambda \frac{z_t}{1 + e^{z_t(\alpha_t + \beta_t)}} + (1 - \lambda) \frac{z_t}{1 + e^{z_t \alpha_t}}, \qquad \beta_t' = \lambda \frac{z_t}{1 + e^{z_t(\alpha_t + \beta_t)}}. \qquad (7)$$

From $z_t > 0$, we see that $\beta_t$ is an increasing function of time, which guarantees $\beta_t > 0$, and

$$\alpha_t' \geq \beta_t' + (1 - \lambda) \frac{z_t}{1 + e^{z_t(\alpha_t + \beta_t)}} = (1 + \frac{1 - \lambda}{\lambda}) \beta_t' = \frac{\beta_t'}{\lambda}.$$

This proves that $\forall t \geq 0, \; \alpha_t \geq \beta_t/\lambda$. We now consider $t^*$ such that $z_{t^*} \alpha_{t^*} = \log(\frac{1 - \delta}{\delta})$. It is the smallest $t$ such that $P_t((x_1, x_2) \in D_1) \geq P_t((x_1, 0) \in D_1) = 1 - \delta$, its existence is guaranteed by $z_t$ and $\alpha_t$ being increasing (we also assume that $t = 0$ does not verify those (in)equalities). We get

$$P_{t^*}((0, x_2) \in D_1) = \frac{1}{1 + e^{-z_{t^*} \beta_{t^*}}} \leq \frac{1}{1 + e^{-\lambda z_{t^*} \alpha_{t^*}}} = \frac{1}{1 + e^{-\lambda \log(\frac{1 - \delta}{\delta})}}.$$

As can be seen in Eq. 7, the presence of $\alpha_t$ – which detects feature $x_1$ – in the denominator of $\beta_t'$ greatly reduces its value, thus preventing the network from learning $x_2$ properly. $\square$

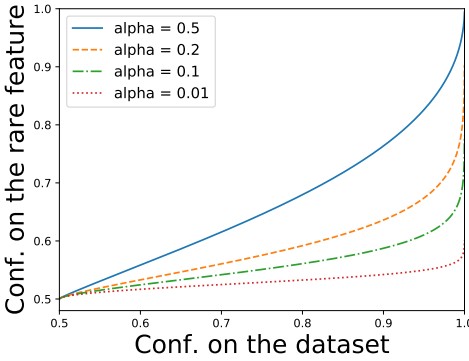

Figure 5: Upper bound on $P_{t^*}((0, x_2) \in D_1)$ as a function of $1 - \delta$ for different values of $\lambda$.

Table 1: Gradient starvation

| | | $\lambda$ | | |
|---|---|---|---|---|
| $\delta$ | 0.5 | 0.2 | 0.1 | 0.01 |
| 99% | 91% | 71% | 61% | 51% |
| 99.99% | 99% | 86% | 72% | 52% |

Table 2: Accuracy on the cats and dogs dataset (Real means the untouched test set)

| Training | Testing | Testing (Real) |
|---|---|---|
| 100% | 100% | 43.2% |

In Fig. 5 *Left.*, we plot for different values of $\lambda$ the confidence of the network when classifying $x_2$ as a function of its confidence on $x_1$ (see App. D. for more details). The gap between the two is very significant: with *e.g.* $\lambda = 0.1$ and $\delta = 10^{-4}$ (Table 1), $P_{t^*}((0, x_2) \in D_1) \leq 72\%$! Even though $x_2$ is exclusively present in $D_1$, and is thus extremely informative, the network is unable to classify it.

**Experiment** To validate those findings empirically, we design an artificial experiment based on the cats and dogs dataset (Kaggle, 2018). We create a very strong, perfectly discriminative feature by making the dog pictures brighter, and the cat pictures darker. We then train a standard deep neural network to classify the modified images and measure its performance on the untouched testing set.

**Results** The results can be seen in Table 2. The network perfectly learns to classify both the train and test modified set, but utterly fails on the real test data. This proves that the handcrafted light feature was learnt by the network, and is used exclusively to classify images. All the features allowing to recognize a cat from a dog are still present in the data, but the low level features (*e.g.* the presence of whiskers, how edges combine to form the shape of the animals and so on) are far less frequent than the light intensity, and thus were not learnt. *The most frequent feature starved all the others.*

## 6 RELATED WORK

The learning dynamics of neural networks have been explored for decades. Baldi & Hornik (1989) studied the energy landscape of linear networks and the fixed point structure of gradient descent

learning in that context. Heskes & Kappen (1993) developed a theory encompassing stochastic gradient descent and parameter dynamics and wrote down their evolution equations in on-line learning. However, those equations are heavily nonlinear and do not have closed-form solutions in the general case. Saxe et al. (2013b) study the case of deep *linear* networks trained with regression. They prove the existence of nonlinear learning phenomena similar to those seen in simulations of nonlinear networks and provide exact solutions to the dynamics of learning in the linear case. Some of our results are an extension of theirs to nonlinear networks. Choromanska et al. (2014); Raghu et al. (2017); Saxe (2015); Yosinski et al. (2014) also focus on neural network dynamics, while Nacson et al. (2018); Xu et al. (2018); Soudry et al. (2017) study the convergence rate of learning on separable data. Arora et al. (2018) prove that overparameterization can lead to faster optimization.

Recent work in the domain of generative adversarial networks (Goodfellow et al., 2014) has shown the resurgence of the hinge loss (Rosasco et al., 2004). In particular, part of the success encountered by Miyato et al. (2018) is due to their use of that specific loss function. Their main contribution however is a spectral normalization technique that produces state-of-the-art results on image generation. Their paper is part of a larger trend focusing on the spectra of neural network weight matrices and their evolution during learning (Vorontsov et al., 2017; Odena et al., 2018; Pennington et al., 2017). It, nevertheless, remains a poorly understood subject.

Zhang et al. (2016) performed some experiments proving that deep neural networks expound a so-called *implicit regularization*. Even though they have the ability to entirely memorize the dataset, they still converge to solutions that generalize well. A variety of explanations for that phenomenon have been advanced: correlation between flatness of minima and generalization (Hochreiter & Schmidhuber, 1997), natural convergence of stochastic gradient descent towards such minima (Kleinberg et al., 2018), built-in hierarchical representations (LeCun et al., 2015), gradient descent naturally protecting against overfitting (Advani & Saxe, 2017), and structure of deep networks biasing learning towards simpler functions (Neyshabur et al., 2014; Perez et al., 2018). Our results from Section 5 suggest that gradient descent indeed has a beneficial effect, but can also hurt in some situations.

## 7 Discussion

In order to obtain closed form solutions for the learning dynamics, we made the extremely simplifying assumption that each class only contains one point. We leave overcoming that limitation to future work. In the spirit of the proof in Section 5 where we considered two datapoints, we might be able to obtain upper and lower bounds on the learning dynamics.

Our comparison between the cross-entropy and the hinge losses reveals fundamental differences. It is noteworthy that the hinge loss is an important ingredient of the recently introduced *spectral normalization* (Miyato et al., 2018). The fast convergence of networks trained with the hinge loss might in part explain its beneficial impact. A deeper analysis of the connections between the two would lead to a better understanding of the performance of the algorithm.

In this paper, we introduce the concept of *gradient starvation* and suggest that it might be a plausible explanation for the generalization abilities of deep neural networks. By focusing most of the learning on the frequent features of the dataset, it makes the network ignore the idiosyncrasies of individual datapoints. That rather desirable property has a downside however: very informative but rare features will not be learnt during training. This strongly limits the ability of the network to transfer to different data distributions where *e.g.* the rare feature exists on its own.

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

APPENDIX A.

A.1  PROOF OF LEMMA 3.1

In this section, we prove the following lemma:

**Lemma 3.1** For any $k \in \{1, 2\}$, $x \in D_k$ and $t \geq 0$, the only non-negative elements of $W_t x$ are the ones with an index $i \in \mathcal{I}_k$. The signs of the coordinates of $Z_t$ remain the same throughout training.

*Proof.* We prove this with a simple induction. The claim is true at $t = 0$ from assumptions (H1-3). Let us assume that at time set $t$, the different parts of the lemma are true. The SGD updates to the weights stemming from a single observation $x \in D_k$ (the extension to a mini-batch is straightforward) with a learning rate $\alpha$ are:

$$\Delta Z_t(x) = \alpha \delta_f(x) \, (W_t x)_+ \qquad\qquad \Delta W_t(x) = \alpha \delta_f(x) \, (Z_t^T \otimes e_k) \, x^T, \qquad (8)$$

where $\otimes$ denotes the element-wise product of two vectors and $e_k \in \mathbb{R}^h$ is the binary vector with ones on the indices from $\mathcal{I}_k$. $\delta_f(x)$ is the gradient of the loss with respect to the pre-sigmoid output of the network $F_t(x)$: $\delta_f(x) = 1_{\{k=1\}} - \sigma(F_t(x))$. The updates to $Z_t$ are positive on its indices belonging to $\mathcal{I}_1$, and negative otherwise, proving the second claim of the lemma. Moving to $W_t$, only its rows and hidden neurons with indices $i \in \mathcal{I}_k$ are modified. For an element $x' \in D$, $W_{t+1} x' = W_t x' + \alpha \delta_f(x)(\sum_{i \in \mathcal{I}_k} z_i) \|x\|'$ where $z_i$ is the $i$-th element of $Z_t$. By induction, $\sum_{i \in \mathcal{I}_k} z_i$ has the same sign as $\delta_f(x)$ (positive on $D_1$, negative on $D_2$). From (H1), $\|x\|'$ is positive (resp. negative) if $x'$ is in $D_k$ (resp. otherwise). The update keeps the $k$-th neuron active (resp. inactive). □

A.2  EXTENSION OF LEMMA 3.1 TO MULTI-CLASS CLASSIFICATION

Let us consider a simple $C$-classification task. $D = \{(x_i, y_i)\}_{1 \leq i \leq n} \subset \mathbb{R}^d \times C$ is our dataset of vectors/labels where $C$ denotes both the set of possible labels and its cardinal depending on the context. The classifier we are training is a simple neural network with one hidden layer of $C$ neurons, a ReLU non-linearity and a softmax $\sigma$. The full function is written as

$$P_t(x) := \sigma(F_t(x)) := \sigma(Z_t(W_t x)_+), \qquad (9)$$

where $W_t$ (resp. $Z_t$) is a $C \times d$ (resp. $C \times C$) weight matrix and $\sigma$ the softmax function. The subscript $+$ denotes the positive part of a real number. $P_t(x)$ is a $C$-vector representing the probability that $x$ belongs to each class. The subscript $t$ denotes the state of the element at time step $t$ of training.

For a given class $k$, we let $D_k$ be the set of vectors belonging to class $k$. We make the following assumptions, with $k \neq k' \in C$ two arbitrary classes

  (H4) For any $x, x' \in D_k$, $x^T x' > 0$. For any $x \in D_k$ and $x' \in D_{k'}$, $x^T x' \leq 0$.

  (H5) For any $x \in D_k$ and $x' \in D \setminus D_k$, $w_0^k x > 0$ and $w_0^k x' \leq 0$, where $w_0^k$ is the $k$-th row of $W_0$.

  (H6) $Z_0$ is initialized randomly to positive numbers on the diagonal, and non-positive elsewhere.

Those assumptions are straightforward extensions of the ones from the main text. We train the classifier using stochastic gradient descent on the cross-entropy loss of our problem. $F_{t+1}$ is the state of the neural network after one SGD update to $F_t$. Our first lemma states that over the course of training, the active neurons of the hidden layer remain the same for each element of the dataset, and the elements of $Z_t$ remain of the same sign.

**Lemma A.1.** *For any $k \in C$, $x \in D_k$ and $t \geq 0$, the only non-negative element of $W_t x$ is its $k$-th element and all diagonal (resp. non-diagonal) elements of $Z_t$ are positive (resp. negative).*

*Proof.* We prove this with a simple induction. The claim is true at $t = 0$ from assumptions (H4-6). Let us assume that at time set $t$, the different parts of the lemma are true. The SGD updates to the weight matrices stemming from a single observation $x \in D_{k^*}$ (the extension to a mini-batch is straightforward) and a learning rate $\alpha$ are:

$$\delta Z_t = \alpha \nabla_y L \, (W_t x)_+^T \qquad\qquad \delta W_t = \alpha (Z_t^T \, \nabla_y L \otimes e_{k^*}) x^T, \qquad (10)$$

where $\otimes$ denotes the element-wise product of two vectors and $e_{k^*}$ the $k^*$ basis vector of $\mathbb{R}^C$. $\nabla_y L$ is the gradient of $\log F_t(x)_{k^*}$ (the cross entropy loss for a sample from class $k^*$) with respect to the output of $Z_t$. One can show that $(\nabla_y L)_{k^*} = 1 - \frac{e^{F_t(x)_{k^*}}}{\sum_j e^{F_t(x)_j}}$ and $(\nabla_y L)_i = -\frac{e^{F_t(x)_i}}{\sum_j e^{F_t(x)_j}}$ for $i \neq k^*$ i.e. $\nabla_y L = e_{k^*} - Y_t(x)$. The update to $Z_t$ is non-negative on the diagonal, and non-positive elsewhere, which proves the second claim of the lemma. As far as $W_t$ is concerned, only its $k^*$-th row is modified. For an element $x' \in D$, $W_{t+1}x' = W_t x' + Kx^T x' e_{k^*}$ where $K$ is the dot product between the $k^*$-th column of $Z_t$ and $\nabla_y L$. By assumptions on the data, $x^T x'$ is positive (resp. negative) if $x'$ is in $D_{k^*}$ (resp. otherwise), so the update keeps the $k^*$-th neuron active (resp. inactive). $\qquad\square$

The proof can be extended to any number of hidden layers $h \geq C$ by modifying assumptions (H5-6) using a partition $\{\mathcal{I}_1, \ldots, \mathcal{I}_C\}$ of $\{1, \ldots, h\}$ similar to the binary classification case. Each set $\mathcal{I}_i$ in the partition describes the neurons active at the initialization of the network for an element $x \in D_i$. The modified assumptions are:

(H5') For any $i \in \mathcal{I}_k$, $x \in D_k$ and $x' \in D \setminus D_k$, $w_0^i x > 0$ and $w_0^i x' \leq 0$.

(H6') For any $i \in \mathcal{I}_k$, $j \notin \mathcal{I}_k$, $(Z_0)_{ki} > 0$ and $(Z_0)_{kj} \leq 0$.

The lemma then translates to those inequalities remaining true at any time $t \geq 0$.

### A.3  Proofs for Theorem 3.2

In this section we develop the proof of Theorem 3.2 of the main text.

*Proof.* We consider an update made using $x \in D_1$. Writing $y_t = w_t x$, our system follows the system of ordinary differential equations

$$y_t' = \frac{x^T x \, z_t}{1 + e^{y_t z_t}}, \qquad\qquad z_t' = \frac{y_t}{1 + e^{y_t z_t}}. \qquad (11)$$

Writing $x^T x = \|x\|^2$, we see that $y_t y_t' = \|x\|^2 z_t z_t'$. The quantity $y_t^2 - \|x\|^2 z_t^2$ is thus an invariant of the problem. With $c := |y_0^2 - \|x\|^2 z_0^2|$, the solutions of Eq. 11 live on hyperbolas of equation

$$\begin{cases} y^2 - \|x\|^2 z^2 = \quad c & \text{if } y_0^2 - \|x\|^2 z_0^2 > 0, \\ y^2 - \|x\|^2 z^2 = -c & \text{if } y_0^2 - \|x\|^2 z_0^2 < 0, \\ y^2 - \|x\|^2 z^2 = \quad 0 & \text{if } y_0^2 - \|x\|^2 z_0^2 = 0. \end{cases} \qquad (12)$$

We start by treating the case of a degenerate hyperbola $c = 0$. We have $\forall t, y_t = \|x\| z_t$ (those quantities are both positive as $x \in D_1$ and (H2-3)). We let $u(t) := z_t w_t x = z_t y_t$ and see by combining the two equations in Eq. 11 that

$$u'(t) = \frac{2\|x\| u(t)}{1 + e^{u(t)}}. \qquad (13)$$

For any $u_f \geq u_0$, let $t = u^{<-1>}(u_f)$ ($u$ is a bijection from $\mathbb{R}^+ \to [u_0, +\infty[$ since its derivative is strictly positive). We have

$$t = \frac{1}{2\|x\|} \int_{u_0}^{u_f} \frac{1 + e^y}{y} dy = \frac{1}{2\|x\|} \left( \log(\frac{u_f}{u_0}) + Ei(u_f) - Ei(u_0) \right), \qquad (14)$$

where $Ei(x) = -\int_{-x}^{\infty} \frac{e^{-u}}{u} du$ is the exponential integral. In the end, with the superscript $<-1>$ denoting the inverse function

$$u_f = u(t) = (\log + Ei)^{<-1>} (2\|x\| t + \log(u_0) + Ei(u_0)). $$

We let $\bar{u}$ denote the function $u(t)$ above for $\|x\| = 1$, the solution for $\|x\| \neq 1$ can easily be inferred by a rescaling of $t$ in $\bar{u}$. For $x \in D_2$, the system of ODEs is

$$y_t' = -\frac{\|x\|^2 \, z_t}{1 + e^{-y_t z_t}}, \qquad\qquad z_t' = -\frac{y_t}{1 + e^{-y_t z_t}}, \qquad (15)$$

the degeneracy assumption becomes $y_0 = -\|x\|z_0$ (we know from (H2) that $y_0 > 0$ and from (H3) that $z_0 < 0$). It can be shown similarly that $v(t) := z_t y_t$ verifies the equation $v'(t) = 2\|x\|v(t)\sigma(v(t))$ with a negative initial condition. In other words, $u$ and $v$ follow symmetric trajectories on the positive/negative real line. Below, $\bar{u}$ and $\bar{v}$ denote those two trajectories for $\|x\| = 1$ and initial conditions $u_0 > 0$ and $v_0 < 0$.

Let us now write $p_1 = |D_1|/|D|$ the fraction of points in the dataset belonging to $D_1$. Because we sample randomly from the dataset, this amounts to sampling $p_1$ (resp. $1 - p_1$) points from $D_1$ (resp. $D_2$) for each time unit during training, ie to rescaling the time axis by $p_1$ for $D_1$ and $1 - p_1$ for $D_2$. Formally, this allows us to quantify the performance of the network at any time $t$

$$\begin{cases} P_t(x \in D_1) = \sigma(\bar{u}(\|x\|p_1 t)) & \text{if } x \in D_1, \\ P_t(x \in D_2) = \sigma(-\bar{v}(\|x\|(1 - p_1)t)) & \text{if } x \in D_2, \end{cases} \tag{16}$$

which concludes the proof for $c = 0$.

From Eq. 14, we also see that

$$t = \frac{1}{2\|x\|} \int_{u_0}^{u_f} \frac{1 + e^y}{y} dy \geq \frac{1}{2\|x\|} \int_{u_0}^{u_f} e^{\frac{y}{2}} dy = \frac{1}{\|x\|}(e^{\frac{u_f}{2}} - e^{\frac{u_0}{2}}), \tag{17}$$

where we used the inequality $\forall y \geq 0, e^{\frac{y}{2}} > y$. It eventually gives us

$$u(t) \leq 2\log(\|x\|t + e^{\frac{u_0}{2}}), \tag{18}$$

a result in line with convergence rates obtained in Soudry et al. (2017).

Let us now study the non-degenerate case. We apply the classic change of coordinates

$$y_t = \sqrt{c}\cosh(\frac{\theta}{2}), \qquad z_t = \frac{\sqrt{c}}{\|x\|}\sinh(\frac{\theta}{2}) \qquad \text{if } y_0^2 > \|x\|^2 z_0^2, \tag{19}$$

$$y_t = \sqrt{c}\sinh(\frac{\theta}{2}), \qquad z_t = \frac{\sqrt{c}}{\|x\|}\cosh(\frac{\theta}{2}) \qquad \text{if } y_0^2 < \|x\|^2 z_0^2. \tag{20}$$

Since $y_t^2 + \|x\|^2 z_t^2 = c\cosh(\theta)$ and $y_t z_t = \frac{c}{2\|x\|}\sinh(\theta)$, we see that

$$(y_t z_t)' = \frac{y_t^2 + \|x\|^2 z_t^2}{1 + e^{y_t z_t}} = \frac{c}{2\|x\|}\cosh(\theta)\theta' = \frac{c\cosh(\theta)}{1 + e^{c\sinh(\theta)/2\|x\|}}, \tag{21}$$

where the first equality used the system of equations Eq. 11. This gives us the dynamics of $\theta$ as

$$\theta' = \frac{2\|x\|}{1 + e^{c\sinh(\theta)/2\|x\|}},$$

with an initial condition $\theta_0 = \cosh^{-1}(\frac{y_0^2 + \|x\|^2 z_0^2}{c})$. For any $\theta_f \geq \theta_0$, we see that $t = \theta^{<-1>}(\theta_f)$ verifies

$$t = \int_{\theta_0}^{\theta_f} \frac{1 + e^{c\sinh(\theta)/2\|x\|}}{2\|x\|} d\theta.$$

There is no closed-form solution for that integral (that we know of). It can however be computed numerically. On Fig. 2 *Right* of the main text, we plot the curves for $y_t$ and $\sigma(z_t y_t)$ for different values of $c$ and $\|x\|$. We obtain a sigmoidal shape similar to previously made empirical observations. We notice in particular that for larger values of $\|x\|$ the function converges faster. $\qquad\square$

## A.4 EXTENSION TO $h$ HIDDEN NEURONS

We now extend the result from Theorem 3.2 to the case with $h$ hidden neurons. Let us still consider an update made on $x \in D_1$. We know from assumptions and by Lemma 3.1 that the only active neurons in the network are indexed by $\mathcal{I}_1$. The network weights follow the evolution equations:

$$(w_t^i)' = \frac{x^T z_t^i}{1 + e^{\sum_{j \in \mathcal{I}_1} z_t^j w_t^j x}}, \qquad (z_t^i)' = \frac{w_t^i x}{e^{\sum_{j \in \mathcal{I}_1} z_t^j w_t^j x}}.$$

We similarly define $y_t^i = w_t^i x$ which brings

$$(y_t^i)' = \frac{x^T x \, z_t^i}{1 + e^{\sum_{j \in \mathcal{I}_1} z_t^j y_t^j}}, \qquad\qquad (z_t^i)' = \frac{y_t^i}{e^{\sum_{j \in \mathcal{I}_1} z_t^j y_t^j}}.$$

The couple $(y_t^i, z_t^i)$ follows the same hyperbolic invariance, defined by a constant $c_i$. In the case where $\forall i \in \mathcal{I}_1$, $c_i = 0$, we see that $u_t^i := z_t^i y_t^i$ verifies

$$(u_t^i)' = \frac{2\|x\| \, u_t^i}{1 + e^{\sum_{j \in \mathcal{I}_1} u_t^j}},$$

and a simple summation on $i$ shows that $u_t := \sum_{i \in \mathcal{I}_1} u_t^i$ follows Eq. 13. The dynamics of the logit in this case are identical to the single active neuron case, the only difference is potentially its initial value.

## A.5 PROOF OF COROLLARY 3.3

**Corollary 3.3** Let $\delta$ be the required accuracy on the classification task (*i.e.* $P_t(x \in D_1) \geq 1 - \delta$ for $x \in D_1$). Under certain assumptions, the times $t_1^*$ and $t_2^*$ required to reach that accuracy for each classifier verify $\frac{t_2^*}{t_1^*} \approx \frac{\|x_1\|}{\|x_2\|}\frac{p}{1-p}$ where $\|x_k\|$ is the norm of vector $\|x_k\|$ from class $k$.

*Proof.* We consider the case $c = 0$. The classification error drops at a rate proportional to $\|x_1\|p$ for $D_1$ and to $\|x_2\|(1 - p)$ for $D_2$. More precisely, let us assume that $u_0 \leq |v_0|$ and look for a classification confidence of $1 - \delta$. We write $u_f = \sigma^{<-1>}(1 - \delta) = \log(1/\delta - 1)$, $t_v = \bar{u}^{<-1>}(|v_0|)$ and

$$t^* = \bar{u}^{<-1>}(u_f) = \frac{1}{2}\left(\log\left(\frac{\log(1/\delta - 1)}{u_0}\right) + Ei(\log(1/\delta - 1)) - Ei(u_0)\right),$$

$t^*$ represents the time taken to reach confidence $\delta$ with an initialization $u_0$, and $t_v$ the time to reach $v_0$ starting in $u_0$. We see that

$$P(x \in D_1) \geq 1 - \delta \iff t \geq t_1^* = \frac{t^*}{\|x_1\|p} \qquad\qquad \text{if } x \in D_1,$$

$$P(x \in D_2) \geq 1 - \delta \iff t \geq t_2^* = \frac{t^* - t_v}{\|x_2\|(1 - p)} \qquad\qquad \text{if } x \in D_2.$$

The ratio between the convergence times reads $\frac{t_2^*}{t_1^*} = \frac{\|x_1\|}{\|x_2\|}\frac{p}{1-p}\left(1 - \frac{t_v}{t^*}\right)$. One sees that if the weight initializations $u_0$ and $v_0$ are close (*i.e.* $t_v$ is small) and the confidence requirement large (*i.e.* $t^*$ is large), the ratio is approximately $\frac{\|x_1\|}{\|x_2\|}\frac{p}{1-p}$. $\qquad\square$

## A.6 TOP-LEFT QUADRANT INITIALIZATION

When the initial conditions of the network verify $y_0 = -\|x\|z_0$, then at all time $t$, $y_t = -\|x\|z_t$. Plugging that equality in Eq. 11 results in

$$u'(t) = \frac{-2\|x\|u(t)}{1 + e^{u(t)}}$$

where again $u(t) = z_t y_t$. This means that the logit is negative and increasing. Let $u_0 < u_f < 0$, we see that the time at which $u$ reaches $u_f$ verifies

$$t = -\frac{1}{2\|x\|}\int_{u_0}^{u_f}\frac{1 + e^y}{y}dy = \frac{1}{2\|x\|}\int_{-u_f}^{-u_0}\frac{1 + e^{-y}}{y}dy \geq \log(-u_0) - \log(-u_f). \qquad (22)$$

$t$ diverges to $+\infty$ as $u_f$ tends to 0 from below. The logit converges to 0 without ever reaching it.

### A.7 Relaxing the single datapoint assumption

One of the major assumptions made in the main text is the fact that each class contains a single element. In this section, we slightly relax it to the case where the points $(\{x_i\}_{1 \leq i \leq m} \subset \mathbb{R}^d)$ in a class are all orthogonal to one another[1] (while still verifying Assumption (H1)). In that case, each presentation of a training vector will only affect $w_t$ in the direction of that specific vector. Let $y_t^i$ denote $w_t x_i$, the unnormalized component of $w_t$ along $x_i$. We consider a batch update on the weights of the neural network. In that case:

$$(y_t^i)' = \frac{\|x_i\|^2 \, z_t}{1 + e^{z_t y_t^i}}, \qquad\qquad (z_t)' = \sum_{i=1}^{m} \frac{y_t^i}{1 + e^{z_t y_t^i}} \, .$$

Assuming that the vectors all have the same norm (denoted $\|x\|$ below) and that the $y_0^i$ are all equal, then that equality remains true at all time (they follow the same update equation). We let $y_t$ denote that value:

$$(y_t)' = \frac{\|x\|^2 \, z_t}{1 + e^{z_t y_t}}, \qquad\qquad (z_t)' = \frac{m y_t}{1 + e^{z_t y_t}} \, .$$

A new invariant appears in those equations: $c := |m y_0^2 - \|x\|^2 z_0^2|$. In the case $c = 0$ (the other case can be treated as above), we obtain the following evolution equation for the logit of any point in the class:

$$u'(t) = \frac{2\sqrt{m}\|x\|u(t)}{1 + e^{u(t)}} \, .$$

We end up with a similar equation than before except for the $\sqrt{m}$ factor, which boosts the convergence speed. However, one should not forget that we are now training on a full batch (*i.e.* on $m$ points) during each unit of time. Performing the same number of updates for the single point class would generate a $m$ factor in the convergence speed of $u$ (one $\sqrt{m}$ factor for each $y$ and $z$ functions). The slower convergence for the more general case can be explained by the fact that each point is making an update on $w_t$ in its own direction. That direction being orthogonal to all others points makes it useless for their classification.

### A.8 Relaxing assumption (H2)

Let us first recall that the assumption states:

(H2) For any $i \in \mathcal{I}_k$, $x \in D_k$ and $x' \notin D_k$, $w_0^i x > 0$ and $w_0^i x' \leq 0$.

We now assume that $h = 2$ and study the evolution of the first row $w_t^1$ of matrix $W_t$, written $w_t$ in the following. We relax assumption (H2) by assuming that there is a point $x_2$ in $D_2$ such that $w_0 x > 0$. And we consider updates to $w_t$ coming from sampling equally $x_1$ from $D_1$ and $x_2$ from $D_2$. The evolution equations can be written as

$$w_t' = \frac{x_1^T \, z_t}{1 + e^{z_t w_t x_1}} - \frac{x_2^T \, z_t}{1 + e^{-z_t w_t x_2}}, \qquad\qquad z_t' = \frac{w_t \, x_1}{1 + e^{z_t w_t x_1}} - \frac{w_t \, x_2}{1 + e^{-z_t w_t x_2}} \, .$$

In order to make the analysis simpler, we assume that $x_1^T x_2 = 0$. We write $\alpha_t = w_t x_1$ and $\beta_t = w_t x_2$. Any component of $w_0$ orthogonal to both $x_1$ and $x_2$ will be untouched by the updates, and does not affect the classification performance of the network. This gives us

$$\alpha_t' = \frac{\|x_1\|^2 z_t}{1 + e^{z_t \alpha_t}}, \qquad \beta_t' = -\frac{\|x_2\|^2 z_t}{1 + e^{-z_t \beta_t}}, \qquad z_t' = \frac{\alpha_t}{1 + e^{z_t \alpha_t}} - \frac{\beta_t}{1 + e^{-z_t \beta_t}} \, . \qquad (23)$$

By assumption, we know that $\alpha_0, \beta_0 > 0$. The system of ODEs (23) is invariant through the transformation $(\alpha_t, \beta_t, z_t) \to (\beta_t, \alpha_t, -z_t)$ so it is sufficient to study the case $z_0 \geq 0$. Let us now state the theorem from the main text.

**Theorem 3.4.** *Letting $c := \alpha_0^2/\|x_1\|^2 + \beta_0^2/\|x_2\|^2 - z_0^2$, the solutions of (23) verify for all $t \geq 0$,*

$\alpha_t^2/\|x_1\|^2 + \beta_t^2/\|x_2\|^2 - z_t^2 = c$. *In other terms, they live on hyperboloids (see Fig. 3 from the main text and Fig. 6).*

---

[1]This implies in particular $m \leq d$.

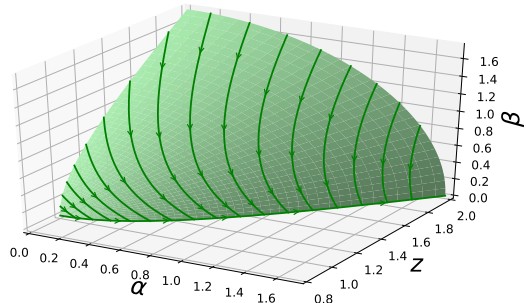

Figure 6: Solutions of the ODE system for different initializations and $c = -1$. Trajectories live on a hyperboloid of two sheets. Any initialization on that surface will result in $\beta_t$ reaching 0, or in other terms in class $D_1$ prevailing. This curve and Fig. 3 from the main text are plotted with $\|x_1\| = \|x_2\| = 1$.

*If $c \leq 0$, $\beta_t$ reaches 0 at some point during training (see Fig. 6). If $c > 0$, there exists a curve $\mathcal{C}_c$ on the hyperboloid such that as $t \to +\infty$, for any initialization $(\alpha_0, \beta_0, z_0) \in \mathcal{C}_c$:*

$$\alpha_t \to \left(\frac{1}{\|x_1\|^2} + \frac{1}{\|x_2\|^2}\right)^{-1/2}\sqrt{c}, \qquad \beta_t \to \left(\frac{1}{\|x_1\|^2} + \frac{1}{\|x_2\|^2}\right)^{-1/2}\sqrt{c}, \qquad z_t \to 0.$$

*That curve defines two regions of the initialization space. In one (colored yellow on Fig. 3 Left), trajectories verify $\alpha_t = 0$ for some t, in the other (colored green) $\beta_t$ reaches 0 at some point.*

*Proof.* It is easy to see from the system of ODEs (23) that $(\alpha_t^2)'/\|x_1\|^2 + (\beta_t^2)'/\|x_2\|^2 - (z_t^2)' = 0$, which directly gives the invariance of $\alpha_t^2/\|x_1\|^2 + \beta_t^2/\|x_2\|^2 - z_t^2$. This implies that the trajectories $(\alpha_t, \beta_t, z_t)$ live on hyperboloids.

If $c < 0$, the hyperboloid has two sheets (see Fig. 6). In particular, the trajectories verify: $z_t^2 = \alpha_t^2/\|x_1\|^2 + \beta_t^2/\|x_2\|^2 - c \geq -c$, which means that $z_t$ is bounded away from 0. As long as $\beta_t \geq 0$, we have $\beta_t' = -\frac{\|x_2\|^2 z_t}{1 + e^{-z_t \beta_t}} \leq \frac{\|x_2\|^2 c}{2}$. This implies that $\beta_t$ will reach 0 in a finite time since it decreases at a rate larger than a strictly positive number. At that point, the ReLU ensures that $\beta_t$ does not evolve anymore, and that $\beta_t$'s contribution to $z_t$ disappears. The evolution equations turn into the ones studied in the previous paragraphs, plotted as the green hyperbola in the plane $\beta = 0$ in Fig. 6.

If $c = 0$, we know that $z_t^2 = \alpha_t^2/\|x_1\|^2 + \beta_t^2/\|x_2\|^2 \geq \alpha_t^2/\|x_1\|^2 \geq \alpha_0^2/\|x_1\|^2$ ($\alpha_t$ increases as long as $z_t$ is positive), so the same argument about $\beta_t$ holds.

If $c > 0$, let us first note that the point $\left(\left(\frac{1}{\|x_1\|^2} + \frac{1}{\|x_2\|^2}\right)^{-1/2}\sqrt{c}, \left(\frac{1}{\|x_1\|^2} + \frac{1}{\|x_2\|^2}\right)^{-1/2}\sqrt{c}, 0\right)$ belongs to the hyperboloid and is stationary (the three derivatives are 0). Classic results on ordinary differential equations (Tenenbaum & Pollard, 1985) then give the result. Finding a closed-form solution to the shape of $\mathcal{C}_c$ is to the best of our knowledge impossible. One can however obtain an approximation by considering a point $\left(\left(\frac{1}{\|x_1\|^2} + \frac{1}{\|x_2\|^2}\right)^{-1/2}\sqrt{c - \epsilon}, \left(\frac{1}{\|x_1\|^2} + \frac{1}{\|x_2\|^2}\right)^{-1/2}\sqrt{c + \epsilon}, 0\right)$ for a small $\epsilon$ and applying finite difference methods to the evolution equations (23) to build the trajectory. $\square$

## APPENDIX B: DEEPER NEURAL NETWORKS

In this section we study the case of a deeper network with $N - 1$ hidden layers. Similarly to above, we can prove the existence of independent modes of learning. To that end, neurons need to be activated in a disjoint manner from one class to the other. Due to the growing complexity of the interactions between the parameters of the network, this requires very strong assumptions on the initialization of

the network and on the shape of the network. Assuming that the network is written as

$$P_t(x) := \sigma(Z_t^T(Z_t^{N-2}\cdots(Z_t^1(W_t x)_+)_+ + \ldots)_+),$$

with $W_t$ an $h \times d$ matrix and for all $1 \leq i \leq N-2$, $Z_t^i$ an $h \times h$ matrix. We maintain assumption (H2) from the main text, and extend (H3) to all $Z_t^i$ by assuming that they are diagonal, and that the $j$-th element of their diagonal is positive if $j \in \mathcal{I}_1$, negative otherwise. To simplify notations, we go back to assuming $h = 2$ and take an update on $x \in D_1$. Only the first elements of every matrix are modified, we write them $z_t^i$ and keep the notations $z_t$, $w_t$ and $y_t$. We can then write the evolution equations:

$$z_t' = \frac{z_t^{N-2}\cdots z_t^1 y_t}{1 + e^{u(t)}}, \qquad (z_t^i)' = \frac{z_t z_t^{N-2}\cdots z_t^{i+1} z_t^{i-1}\cdots y_t}{1 + e^{u(t)}}, \qquad y_t' = \frac{\|x\|^2 z_t z_t^{N-2}\cdots z_t^1}{1 + e^{u(t)}},$$

with $u(t) = z_t \, \Pi z_t^i \, y_t$. Assuming that $z_0 = z_0^1 = \ldots = z_0^{N-1} = \frac{y_0}{\|x\|}$, we see that those equals remain true throughout training. This gives us

$$z_t' = \frac{(z_t)^{N-1}\|x\|}{1 + e^{u(t)}}, \qquad u(t) = (z_t)^N\|x\|, \qquad u'(t) = N(z_t)^{N-1}z_t'\|x\|.$$

Combining those equations gives us the ODE verified by the logit of our system

$$u'(t) = \frac{N\|x\|^{2/N} u^{2-2/N}(t)}{1 + e^{u(t)}}. \tag{24}$$

The solution of that equation for $N = 4$ and $N = 8$ can be found on Fig. 7. Here too, a sigmoidal shape appears during the learning process. The effect of $\|x\|$ reduces as $N$ grows due to the power $2/N$, however, larger values still converge faster (*e.g.* the blue and yellow curves). Additionally, as noted in Saxe et al. (2013b) for linear networks: *the deeper the network, the faster the learning*. This fact is studied in more details in Arora et al. (2018) where depth is shown to accelerate convergence in some cases.

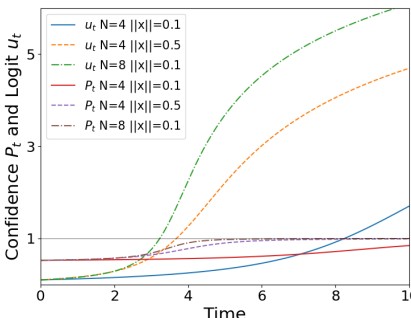

Figure 7: Logit $u(t)$ and confidence $P_t$ for different number of layers and values of $\|x\|$.

## APPENDIX C: HINGE LOSS

### C.1 PROOF OF THEOREM 4.1

In this section we prove Theorem 4.1 from the main text on the dynamics of learning in the case of the Hinge loss:

$$L_H(W_t, Z_t; x) = \max(0, 1 - Z_t^T(W_t x)_+ \cdot (\mathbb{1}_{x \in D_1} - \mathbb{1}_{x \in D_2})).$$

Let us consider updates made after observing a point $x \in D_1$, the converse can be treated similarly with a simple change of sign. The system of ordinary differential equations verified by the parameters of our network is:

$$y_t' = \|x\|^2 \, z_t, \qquad z_t' = y_t. \tag{25}$$

The same relation between $y_t$ and $z_t$ appears in those equations than in the cross-entropy case. Defining $c$ similarly, we have $u'(t) = 2\|x\|u(t)$ in the case where $c = 0$, leading to $u(t) = u_0 e^{2\|x\|t}$. When $c \neq 0$, the same change of variables can be applied and leads to

$$y_t = \sqrt{c}\cosh(\frac{\theta_0}{2} + \|x\|t)\,, \qquad z_t = \frac{\sqrt{c}}{\|x\|}\sinh(\frac{\theta_0}{2} + \|x\|t)\,, \qquad u_t = \frac{c}{2\|x\|}\sinh(\theta_0 + 2\|x\|t)\,, \quad (26)$$

with $\theta_0 = \cosh^{-1}(\frac{y_0^2 + \|x\|^2 z_0^2}{c})$. Those equations are only valid until $u_t$ reaches $1$[2]. At that point learning stops, the network has converged. If the initialization is such that that condition is already verified, then the weights will not change as they already solve the task. The learning curves along with the initialization diagram can be found in Fig. 8. We notice a *hard* sigmoidal shape, corresponding to learning stopping when $u_t$ reaches $1$.

## C.2 GENERAL TREATMENT

Let us now consider the general case of a class containing an arbitrary number of points $D_1 = \{x_i\}_{1 \le i \le m} \subset \mathbb{R}^d$. We consider the case of updates done in full batches (standard gradient descent in other words). In that case, we see that the network obeys the following dynamics:

$$w'_t = z_t \sum_{i=1}^m x_i^T\,, \qquad\qquad z'_t = w_t \sum_{i=1}^m x_i^T\,.$$

Letting $X = \sum_{i=1}^m x_i$ denote the sum of all the datapoints in that class, we see that those dynamics boil down to our previous treatment for a single point. The same cases appear, depending on the value of $c := |(w_0 X)^2 - \|X\|^2 z_0^2|$. We explicitly treat the $c > 0$ case. Following the methods above, we see that: $w_t = y_t \frac{X^T}{\|X\|^2} + w_0^\perp$ where $y_t = \sqrt{c}\cosh(\frac{\theta_0}{2} + \|X\|t)$ and $w_0^T$ is the component of $w_0$ orthogonal to $X$ (and thus unchanged during training). With $z_t$ and $u_t$ defined as above (with $X$ instead of $x$), an arbitrary example $x$ is then classified as

$$P(x \in D_1) = z_t y_t \frac{X^T x}{\|X\|^2} + z_t w_0^\perp x = u_t \frac{X^T x}{\|X\|^2} + z_t w_0^\perp x\,.$$

## APPENDIX D: GRADIENT STARVATION

In this section, we prove a relaxed version of Theorem 5.1 from the main text:

**Theorem D.2.** *Let $\delta$ be our confidence requirement on class $D_1$ i.e. the training stops as soon as $\forall x \in D_1$, $P_t(x \in D_1) \ge 1 - \delta$. Let $t^*$ denote that instant i.e. $z_{t^*}\alpha_{t^*} = \log(\frac{1-\delta}{\delta})$. If $\beta_0 < 0$, the inequality (9) from the main text is valid. Otherwise, with $w_0 = (\alpha_0 x_1, \beta_0 x_2) + (x_1^\perp, x_2^\perp)$, we have*

$$P_{t^*}((0, x_2) \in D_1) \le \frac{1}{1 + e^{-\lambda \log(\frac{1-\delta}{\delta}) - z_{t^*}(\beta_0 - \alpha\alpha_0)}}\,.$$

*Proof.* We start with the $\beta_0 < 0$ case. If $\beta_{t^*}$ is negative, the result from the main text clearly holds. Otherwise, there exists $\tilde{t} < t^*$ such that $\beta_{\tilde{t}} = 0$ ($\beta_t$ is increasing). The proof of Theorem 5.1 from the main text can then directly be applied to $[\tilde{t}, t^*]$. If $\beta_0 > 0$, the inequality on $\alpha'_t$ and $\beta'_t$ holds and gives $\beta_t \le \beta_0 + (\alpha_t - \alpha_0)\lambda$. Plugging it into $P_{t^*}((0, x_2) \in D_1)$ concludes our proof. $\square$

In the main text, we assume that $\beta_0 - \alpha\alpha_0 < 0$ and obtain a bound on the confidence which is independent from $\alpha_0$ and $\beta_0$. Using that bound allows to obtain Fig. 9, but is partly unfair as the initialization of the network is already favoring the strong feature.

However, we note that under small random initialization $z_{t^*}$ and $\alpha_{t^*}$ are of the same order of magnitude and $(\beta_0 - \alpha\alpha_0)$ is very small compared to $\log(\frac{1-\delta}{\delta})$. The additional term in the denominator thus has a limited effect on the exponential, gradient starvation is still happening (a fact confirmed

---

[2]We are considering class $\{1\}$ here, but the equivalent can be proven for class $\{-1\}$.

by the experiment on the cats and dogs dataset). In the main text, Fig. 5 plots the upper bound for a *fair initialization* $\alpha_0 = \beta_0 = 0.1$ (in that case, we need to assume that $z_{t^*} = \alpha_{t^*}$).

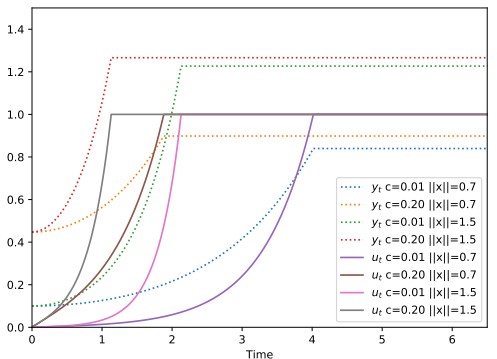

Figure 8: Solution of Eq.26.

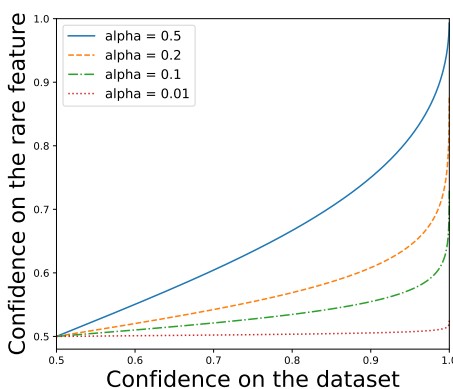

Figure 9: Upper bound on $P_{t^*}((0, x_2) \in D_1)$ as a function of $1 - \delta$ for different values of $\lambda$.

## APPENDIX E: EXPERIMENTAL DETAILS

### E.1    MIXTURE OF GAUSSIAN EXPERIMENT

In this experiment, the data is constructed using eight independent Gaussian distributions around a unit circle. The variance of each Gaussian is chosen such that all eight modes of the data are separated by regions of low data probability, but still contain a reasonable amount of variance. This simple experiment resembles multi-modal datasets. Although this task might seem simple, in practice many generative adversarial networks fail to capture all the modes. This problem is generally known as *mode collapse*.

As shown in the main text, using the hinge loss instead of the common binary cross-entropy loss alleviates the problem significantly. The architectures used for the generator and discriminator both consist of four hidden layers where each layer has 256 hidden units. As a common choice, a ReLU is used as the non-linearity function for hidden units. The length of the noise input vector is 128. The Adam optimizer (Kingma & Ba, 2014) was applied during training with $\alpha = 10^{-4}$, $\beta_1 = 0.5$ and $\beta_2 = 0.9$. The PyTorch framework (Paszke et al., 2017) was used to conduct the experiment.

### E.2    DOGS VS. CATS CLASSIFICATION WITH LIGHT EFFECT

For the purpose of highlighting the fact that *the most frequent feature starved all the others*, we conducted an experiment on a classification task. We modified the cats and dogs dataset (Kaggle, 2018) by setting the cats images to be lighter than the dogs images. To do so, each pixel in a cat image is scaled to be between 0 and 127 while each pixel in a dog image is scaled to be between 128 and 255. The dataset consists of 12500 images of each class. The classifier has an architecture similar to VGG16 (Simonyan & Zisserman, 2014). In order to isolate the effect of the induced bias, no regularization was applied. The Adam optimizer was applied here as well during training with $\alpha = 10^{-4}$, $\beta_1 = 0.9$ and $\beta_2 = 0.99$. The PyTorch framework was used to conduct the experiment.

