# OpenReview forum: "Convergence Properties of Deep Neural Networks on Separable Data"
_ICLR.cc/2019/Conference_

### Official Review · AnonReviewer3 · 2018-11-02
**The paper is concerned with a special case of deep learning (binary classification subject to strong assumptions) and tries to establish a mixed bag of results related to what is called "learning dynamics" of deep learning.**

**Rating:** 5
**Confidence:** 3

**Review:**

The underlying motivation for the paper is really interesting and cuts straight to the heart of Deep Learning and strives to unravel the key understanding that we are still to a large extent missing.

When it comes to clarity and organization I find the paper a bit "messy" in that it is a collection of quite a few findings on the very specific topic of binary classification with quite strong assumptions. Especially given the very specific nature of the topic I miss a strong and clear path through the paper. Unfortunately the paper leaves me with the distinct feeling that there are still a lot of work needed to be able to tell the story about the problem under study. Having said that the paper does contain several individual findings. Having said that I find the ideas leading up to what the authors refers to as "gradient starvation" to be really interesting and that would be a great clear idea to focus on.

A few concrete questions/comments:
Can you explain somewhere exactly what you mean when you say "learning dynamics of deep learning"? Given the specific nature of the results presented in the paper it would be nice to be precise also when it comes to the overall topic under study.

Given the very specific nature of the topic treated in the paper I find the title of the paper largely misleading. The title claims way more than what is actually delivered in the paper, despite the fact that the authors have put in an "On" in the beginning of the title.

In Corollary 3.3. you characterize the convergence speed in a nice way, but I am missing the link to the behaviors observed empirically in e.g. Fig. 2. What am I missing?

The final sentence in Section 2 is highly speculative and I find this hard to believe without solid backing. The sentence reads "... and helps develop intuitions about behaviors observed in more general settings." Given the restrictive nature of your set-up I find it very hard to believe that this extends to more general settings.

Tiny detail: The axes of several of the plots given in the paper mis the lables which makes it hard to read. Straightforward to fix, but worth mentioning nevertheless.

---

> ### Author Response · Authors · 2018-11-22
> **Reply to your questions/comments**
>
> Thank you for your review. Below we attempt to answer your concerns. We also want to point out that we have added some insights/results relaxing one of our main assumptions in Section 3.4 of the latest version of the paper.  For more details, please see the comment above entitled: “Relaxing Assumption (H2)”.
>
> The path we attempted to draw through the paper aims at the evolution of a nonlinear neural network’s classification performance throughout its training and at the factors that influence it: from the norm of the input to the type of loss used for learning or the frequency of features present in the training data. Our framework is able to establish properties on the behavior/convergence of certain classifiers during their training on separable data. Those insights match some observations made by machine learning practitioners, in particular about the sigmoidal shape of learning metrics or the efficiency of the hinge loss on certain tasks.
>
> We have added an explanation of what we mean by “learning dynamics of deep learning” in the last paragraph of the first page. It usually refers to the evolution of weights and outputs of neural networks throughout training. For instance, the work by Saxe et al in 2013 is entitled “Exact solutions to the nonlinear dynamics of learning in deep linear neural networks”. We based our title on that paper since it extends some of its results to nonlinear neural networks.
>
> We understand your concern and have made the title more specific. Tentatively, we chose: “Convergence Properties of Deep Neural Networks on Separable Data”.
>
> Let us assume for simplicity that in Corollary 3.3, p = 0.5 (ie that the classes are balanced) and that ||x_1|| = 1, ||x_2|| = 0.5. Then the confidence of the network on those classes corresponds to the red and dashed purple curves of Fig. 2. Right. In particular, we see that reaching any level of confidence takes approximately twice as much time on class 2 (red curve) than on class 1 (dashed purple curve). That is effectively what the corollary is expressing.
>
> We have edited the corresponding sentence to make it less assertive.
>
> We have added the missing labels in the latest version of the paper. Thank you for pointing it out.

---

### Official Review · AnonReviewer1 · 2018-11-02
**Good starting point to analyze learning of non-linear deep nets, but assumptions are too strong**

**Rating:** 5
**Confidence:** 4

**Review:**

The authors study properties of the learning behavior of non-linear (ReLu) neural networks. In particular, their main focus is on binary classification for the linear-separable case, when optimization is done using gradient descent minimizing either binary entropy or hinge loss.

There are 3 main results in the paper:
1) During learning, each neuron only activates on data points of one class: hence (due to ReLu), each neuron only updates its weights when seeing data points from that class. The authors refer to this property as "Independent modes of learning", suggesting that the learning of parameters of the network is decoupled between the two classes.
2) The classification error, with respect to the number of iterations of gradient descent, exhibits a sigmoidal shape: slow improvement at the beginning, followed by a period of fast improvement, followed by another plateau.
3) Most frequent features, if discriminative, can prevent learning of other, less frequent, features.

Apart from the assumption H1 of linear separability of the data (which I don't mind), the results require very strong assumptions, in particular hypothesis H2 stating "at the beginning of training data points from different classes do not activate the same neurons".

Even for a shallow net, the authors are essentially assuming that the first layer of weights W is such that each row w is already a hyperplane separating the two classes after initialization (wx > 0 for all x belonging to one class and wx' < 0 for x' in the other class). In other words, at initialization, the first layer is already correctly classifying all data points. This is of course an extremely stringent assumption that doesn't hold in practice (eg, the probability of such an initialization shrinks to zero exponentially in the number of dimensions and in the number of neurons).

Because of this concern, I believe the results in the paper can only really characterize the learning close to convergence, since the network is already able to provide correct classification.

Pros:
 - Authors consider a non-linear (ReLu) neural network, as opposed to the analysis of Save et al which only considers linear nets.
 - The fundamentally different behavior between Hinge and binary entropy loss is interesting, and worth analyzing further.
 - Sigmoidal shape of classification error as a function of number of iterations is inline with what is seen in practice. However, I believe the assumptions needed to show this point force the analysis to only characterize learning close to convergence.

Minor Cons (apart from major concern above):
 - Theorem 3.2: "[...] converges at a speed proportional to [...]". Isn't \bar{u}_t logarithmic (non-linear) in t?
 - Theorem 3.2: Even if strong, I don't mind the assumption on a dataset merely consisting of two (weighted) data points. I would suggest to simulate this case without putting any condition on the initialization of the weights (ie, without assumptions H1-H2), and compare the empirical shape of the classification error with the one you obtain analytically in Figure 2 Right.
- Theorem 3.2 Interpretation: unfinished sentence "We can characterize the convergence speeds more quantitatively with the"
- Theorem 4.1: Can you give an intuition or lower/upper bounds for u(t) for the Hinge case, to make evident its difference from the binary entropy case (where u(t) ~ log(t))
- Gradient starvation, Kaggle experiment: I'm not too convinced about the novelty/usefulness of this result. In the end, even a decision tree stump would stop growing after learning the dark/light feature as a discriminator. What I'm trying to say is that "gradient starvation" is a more general problem that really doesn't have to do with gradient descent. Also, the fact that the accuracy on the Kaggle non-doctored test set is low is simply because the test set is not coming from the same distribution of the training set.

---

> ### Author Response · Authors · 2018-11-22
> **Some additional results and improvements**
>
> We thank you for your thorough review, which has undoubtedly helped improve the paper.
>
> First, we agree that assumption (H2) is restrictive and have added some insights/results relaxing it in Section 3.4 in the latest version of the paper. For more details, please see the comment above entitled: “Relaxing Assumption (H2)”.
> Nevertheless, we wish to emphasize that even under Assumption (H2), learning can still fail. Fig 2. Left and Section 3.3 show that any initialization in the top left red region will lead (after a finite number of updates) to a confidence of 0.5 on the corresponding class. The network does not provide correct classification at the end of training even though it does at the beginning.
>
> Here are responses to your other concerns:
>    - Indeed, our intent in the statement of Theorem 3.2 was to describe the scaling of the solution with respect to those two quantities, but it can be misinterpreted. We have clarified it in the new version of the paper.
>    - We have run that experiment and included it in Fig 3. Right among our other recent findings.
>    - Corrected in the new version.
>    - We have added a line in the last paragraph of Section 4 stating that for the Hinge loss, u(t) grows exponentially in t.
>    - We agree that the observed phenomenon can appear in other machine learning methods and is not specific to gradient descent. However, in the case of deep neural networks, it is the prevalence of certain gradient directions that determine the final classifier. Our results suggests that models converge to solutions that privilege the “simplest” explanation, in an Occam’s razor fashion, which provides an explanation to the “implicit generalization” of deep nets characterized by Zhang et al.
>    Our Kaggle experiment’s aim is to emphasize potential failure modes of current architectures/algorithms (one can think of a self-driving car trained on a road with clear lane markings and operating on a road without such markings). The ability to transfer knowledge to test sets coming from a different distribution is key to building more intelligent and robust systems.

---

### Official Review · AnonReviewer2 · 2018-11-03
**Nice insights with too strong assumptions**

**Rating:** 5
**Confidence:** 4

**Review:**

The authors study the learning dynamics of deep neural networks, which is of fundamental importance but lacks understanding. The authors study several dynamics like activation independence, gradient starvation, which gives new insights. However, the assumption is too strong.

There are two main results in the paper:
1) Through learning, the neurons activates of one class.
2) The classification error, with respect to the number of iterations of gradient descent, exhibits a sigmoidal shape.

However, there are two strong assumptions: 1. the two data are perfectly separable by linear classifier. 2.  H2 assumes "at the beginning of training data points from different classes do not activate the same neurons". This is a very strong initial assumption, I am not sure how likely this assumption would be satisfied. It sounds to me this assumption implicitly suggests that the algorithm is already ALMOST CONVERGENT.

If this assumption cannot be weakened, I don't think the paper can be accepted.

---

> ### Author Response · Authors · 2018-11-22
> **Improvements and clarifications**
>
> Thank you for your review.
>
> We agree that assumption (H2) is very restrictive and have added some results relaxing it in Section 3.4 in the latest version of the paper. Please see the comment above entitled: “Relaxing Assumption (H2)” for more details.
> However, it it worth pointing that even under Assumption (H2), learning does not necessarily converge. As shown in Fig 2. Left and Section 3.3, any initialization in the top left red region will fail to solve the problem. In that case, the confidence on the corresponding class will be 0.5 after a finite number of updates.
>
> As far as assumption (H1) is concerned, it is very classic in deep learning theory (see for instance [1,2,3]) and we have not been able to relax it.
> [1] T. Laurent and J. von Brecht. Deep linear networks with arbitrary loss: All local minima are global. ICML 2018
> [2] Z. Liao and R. Couillet. The dynamics of learning: A random matrix approach. ICML 2018.
> [3] S. Arora et al. On the Optimization of Deep Networks: Implicit Acceleration by Overparameterization. ICML 2018.

---

### Author Response · Authors · 2018-11-22
**Relaxing Assumption (H2)**

As pointed out by two reviewers, Assumption (H2) is particularly restrictive and equivalent to assuming that at initialization the network already separates the two classes.

It is worth pointing out that even under such an assumption, there exists a non-zero measure region in the space of initializations where the network is originally able to separate the classes and still eventually fails to converge. We have emphasized that point further in a small paragraph at the end of section 3.3 and in Figure 3 Right.

Nevertheless, we have been able to relax to some extent assumption (H2) and added a section 3.4 in the latest version of the paper (we also moved the section on deeper neural networks to the Appendix).
In short, assuming that both classes activate the same neuron at time t=0:
   1/ We were able to show that eventually one of the class will stop activating said neuron (in which case learning reaches the regime already studied in the original paper).
   2/ We characterized the surface separating the set of initializations in which class 1 “wins over” the neuron from the set where class 2 does (shown in Figure 3 Left).
   3/ We conducted some simulations confirming that even when no analytical solution exists to the problem, the curves still present sigmoidal shapes (Figure 3 Right). In light of our recent extension, it can be interpreted as follows: in the small initialization regime, the competition over the neuron between classes happens in a region of parameter space with small norm. This entails marginal changes in the confidence of the network (which remains close to 0.5), i.e. a plateau on the confidence curve. As soon as one class prevails over the other, the original analytical results from the papers apply, and the sigmoidal shape arises.

Overall, we believe that those new findings significantly strengthen the paper and thank the reviewers for pushing us in that direction.

---

### Author Response · Authors · 2018-11-22
**New Title**

A reviewer pointed out the fact that the title "On the Learning Dynamics of Deep Neural Networks" is too broad for the content of the paper. We acknowledge that remark and have modified it to "Convergence Properties of Deep Neural Networks on Separable Data" which we think is more adapted.

---

### Meta-Review · Area_Chair1 · 2018-12-14

**Confidence:** 4
**Recommendation:** Reject

**Metareview:**

The manuscript proposes to analyze the learning dynamics of deep networks with separable data. A variety of results are provided under various assumptions.

The reviewers and AC note the assumptions required for the analysis are quite strong, and perhaps too strong to provide useful insight into real problems. Reviewers also cite issues with writing and the breadth of the title (this was much improved after rebuttal).